# Effects of Maternal Supplementation with Rare Earth Elements during Late Gestation and Lactation on Performances, Health, and Fecal Microbiota of the Sows and Their Offspring

**DOI:** 10.3390/ani9100738

**Published:** 2019-09-28

**Authors:** Yi Xiong, Jiaman Pang, Liangkang Lv, Yujun Wu, Na Li, Shimeng Huang, Zhi Feng, Ying Ren, Junjun Wang

**Affiliations:** 1Hubei Key Laboratory of Animal Nutrition and Feed Science, Wuhan Polytechnic University, Wuhan 430023, China; ctt20180228@163.com (Y.X.); liangkanglvling@163.com (L.L.); fengzhi19910906@163.com (Z.F.); 2State Key Laboratory of Animal Nutrition, College of Animal Science and Technology, China Agricultural University, Beijing 100193, China; pangjm@cau.edu.cn (J.P.); yujun@cau.edu.cn (Y.W.); swunln@163.com (N.L.); shimengh@cau.edu.cn (S.H.)

**Keywords:** REE, performances, fecal microbiota, oxidative stress, sow, offspring

## Abstract

**Simple Summary:**

The immunological and metabolic status of breeding sows directly affect the overall productivity of porcine operations. Especially, maternal health status during the transition from gestation to lactation are important in maintaining health and growth of the suckling piglets. Rare earth elements (REEs) have been considered as a promising natural feed additive and been reported to exert their activity locally within the gastrointestinal tract, including effects on the bacterial microflora and on nutrient utilization. The present study was conducted to explore the effects of dietary maternal REE supplementation during late gestation and lactation on sows and their offspring. After the experiment, we found that maternal REE addition enhanced antioxidant activity and immunity of sows and their suckling piglets. At the same time, REE supplementation during perinatal period improved the reproductivity of the sows as well as the growth of their offspring. Besides, maternal REEs supply altered the intestinal microbiota community and composition of sows as well as their offspring, and Spearman correlation analysis shows that fecal bacteria are associated with the antioxidase, inflammatory factors of the sows and offspring as well as average daily gain of the suckling piglets. In addition, our results suggested that REE supplementation during both gestation and lactation are more beneficial to sows and their offspring than supplementation during only late gestation. This paper holds promise in providing efficient feeding strategies in swine production.

**Abstract:**

The study was conducted to investigate the effects of maternal supplementation with rare earth elements (REEs) on sows and their offspring. During late gestation, 120 multiparous sows were divided randomly into the control group (Basal diet) and REE-G group (Basal diet supplemented with 200 mg REE/kg). After delivery, REE-G group was further divided into two groups: REE-L- (Change to basal diet during lactation) and REE-L+ group (REE diet all the time). Our results showed that maternal REE supplementation improved the antioxidant and immunity of sows and piglets. Additionally, REE supply during late gestation significantly decreased the coefficient of within-litter variation (CV) in birth weight and increased the weaning weights and the average daily gain (ADG) of piglets. During lactation, the insulin-like growth factor-1 (IGF-1) levels in piglets of REE-L+ group were higher, while no difference between REE-L- and the control group. More beneficial bacteria (*Christensenellaceae* and *Ruminococcaceae*) were found in the REE-L+ group while some opportunistic pathogens (*Proteobacteria* and *Campylobacter*) were relatively suppressed. Fecal microbiota showed correlation with antioxidase, inflammatory factors, and average daily gain (ADG). Collectively, our findings indicated that REEs added in both gestation and lactation was more conducive to establish a healthier status for sows and their offspring.

## 1. Introduction

Maternal nutrition and health condition during the transition from late gestation to lactation are not only essential for the sows themselves, but also for the neonatal suckling piglets [1]. Previous study indicates that during the perinatal period, mothers are easier to experience aggravated oxidative stress and inflammatory responses [2]. Indeed, gestational sows exhibit notable changes in gut microbiome and stress responses would increase [3] with any increase in the systemic exposure to microorganisms of maternal origin. The gut microbiota plays an important role in nutrient metabolism and immune system in the host, moreover intestinal microflora changes may directly influence the maternal pregnancy-associated metabolic alterations [4].

Rare earth elements (REEs) include the lanthanides lanthanum (La), Cerium (Ce), and another 15 elements, which are extensively applied in agriculture and other fields [5]. REEs have long been used as a practical feed additive (even an alternative to antibiotics) without residue or safety problem in animal production [6]. Up until now, there are many literatures concerning the performance enhancing effects of REEs for pigs at different stages of life [7,8,9,10]. Better utilization of nutrients was also indicated after feeding trials were performed on pigs and broilers [11,12], in which higher body weight gain was observed along with increased feed conversion ratio. Apart from the growth promoting effects, functions as antioxidant and cellular defense enhancement have been proposed [13]. In addition, the antibacterial activities of Ce^3+^ to *E. coli* cells [14] and the suppressive effect of La on inflammatory response [15] were both found. Furthermore, it was also found that supplementing LaCl_3_ has an impact on rumen microbial flora (increased the relative abundance of *F. succinogenes* and decreased *R. flavefaciens*). Additionally, REE supplementation also has certain effects on the growth and microflora of pigs [11,16,17]. A previous study reported in vivo preferential antimicrobial action of REEs against Gram-negative bacteria despite little effective influence of REEs on the fecal microbiota of nine-week-old piglets are monitored using polymerase chain reaction-denaturing gradient gel electrophoresis (PCR-DGGE) analysis [11]. Importantly, the underdevelopment of intestinal microbiota has the potential to give rise to disorders. Thus, knowledge of the interactions of the microbiota with diets is essential for advancement in modulating and improving livestock health status.

However, to date, the effects of dietary maternal REE supply during perinatal period on sows and their offspring, and the experimental evidence of the relationship between the gut microbiota, antioxidant capacity, immune status and growth are still absent. To test the hypothesis that REEs are beneficial to the performance and health of sows and even affect their offspring (longitudinal progression in time), we evaluate the effects of maternal REE supplementation during late gestation and lactation on reproductive performances of the sows, plasma biochemical changes and fecal microbiota of the sows and their piglets, as well as the growth performances of piglets. 

## 2. Materials and Methods

### 2.1. Animals and Experimental Treatments 

One hundred and twenty multiparous sows (Landrace × Yorkshire, 3–5 parturition) were selected. The experiment was carried out in two phases, phase one is from late gestation to delivery (G90–G114), sows were comprehensively and equally distributed into two groups: the control group (grain-based diet, basal diet) (*n* = 60) and the REE-G group (the basal diet supplemented with 200 mg REE mixture/kg, REE diet) (*n* = 60). Phase two is from delivery to weaning (L1–L21), there were three groups in this stage: control (*n* = 30), REE-L- (*n* = 30) and REE-L+ (*n* = 30) group. During lactation, sows from REE-G group were further assigned into REE-L-group (REE diet during late gestation + basal diet during lactation) and REE-L+ group (REE diet throughout the late gestation and the lactation). Thirty samples were randomly selected from the control group of phase one using the random number generator in SPSS (SPSS v. 22.0 for Windows; SPSS Inc., Chicago, IL, USA) to keep the samples numerically balanced between control and treatment groups. And the control group were continued to be fed a basal diet during the lactation. Complete diets were formulated to provide all nutrients at or above requirement and were shown in Table 1. As REE sources, an REE mixture containing 5.72% of La, 3.26% of Ce and other carrier components (diatomite) as chelating agent were used. All REE mixture samples were obtained from Shenzhen SQA Industrial Co., Ltd. in Guangdong, China. This experiment was approved by the Animal Care and Use Committee of Wuhan Polytechnic University (WHPU20140608-1, Wuhan, China).

### 2.2. Litter Performance Measurement

At parturition, the individual birth weight was recorded to calculate the total birth weight per litter and the average birth weight for the live piglets as well as the coefficient for within-litter birth weight variation (CV is the ratio of standard deviation of body weight to average body weight of newborn piglets at birth). Also, total litter size and born alive were recorded. On day 21 of lactation, weights of weaned piglets were measured to calculate the average daily gain (ADG) during the neonatal stage.

### 2.3. Plasma and Feces Collection

On day 114 of gestation, eight sows per group were selected at random and the blood samples (10 mL) were collected via anterior vena cava into the heparin sodium anticoagulation tube. On day 21 of lactation, eight sows and eight piglets per group (one piglet per litter corresponding to the sow) were randomly selected and the blood samples of sows (10 mL) and piglets (5 mL) were collected in the same way. Plasma samples were then obtained by centrifuging the blood samples at 3000× *g* for 10 min at 4 °C and were immediately stored at −80 °C until further analysis. At the same time, fresh feces of the lactating sows and the weaning piglets were collected and stored at −20 °C until assay.

### 2.4. Analysis of Antioxidase and Inflammatory Cytokines of Plasma Samples

The contents of total-antioxidant capacity (T-AOC), catalase (CAT), total superoxide dismutase (T-SOD), glutathione peroxidase (GSH-Px) and malondialdehyde (MDA) were tested using the commercial assay kits according to the manufacturers’ instructions (Nanjing Jiancheng Bioengineering Institute, Nanjing, China). The concentrations of interleukin-10 (IL-10), interleukin-1β (IL-1β), TNF-α were measured with the commercial porcine ELISA kit (Elabscience Biotechnology Co., Ltd., Wuhan, China) according to the manufacturer’s instructions.

### 2.5. Fecal Microbial Analysis

Microbial DNA was extracted from the feces samples using the E.Z.N.A^®^ soil DNA Kit (Omega Bio-tek, Norcross, GA, USA) according to manufacturer’s protocols. The final DNA concentration and purification were determined by NanoDrop 2000 UV-Vis spectrophotometer (Thermo Scientific, Wilmington, DE, USA), and DNA quality was checked by 1% agarose gel electrophoresis. The V3–V4 hypervariable regions of the bacteria 16S rRNA gene were amplified with primers 338F (5′-ACTCCTACGGGAGGCAGCAG-3′) and 806R (5′-GGACTACHVGGGTWTCTAAT-3′) by thermocycler PCR system (GeneAmp 9700, ABI, Carlsbad, CA, USA) [18]. The PCR reactions were conducted using the following program: 3 min of denaturation at 95 °C, 27 cycles of 30 s at 95 °C, 30 s for annealing at 55 °C, and 45 s for elongation at 72 °C, and a final extension at 72 °C for 10 min. 

Purified amplicons were pooled in equimolar and paired-end sequenced on the Illumina MiSeq platform (Illumina, San Diego, CA, USA) according to the standard protocols by Majorbio Bio-Pharm Technology Co. Ltd. (Shanghai, China). 

Raw fastq files were demultiplexed, quality-filtered by Trimmomatic and merged by FLASH with the following criteria: (i) The reads were truncated at any site receiving an average quality score <20 over a 50 bp sliding window. (ii) Primers were matched with an allowance of 2 nucleotides mismatching, and reads containing ambiguous bases were removed. (iii) Sequences with overlap longer than 10 bp were assembled according to their overlap sequence. Operational taxonomic units (OTUs) were clustered using UPARSE (version 7.1) with a cutoff of 97% similarity and chimeric sequences were identified and removed using UCHIME. The taxonomy of each 16S rRNA gene sequence was analyzed by ribosomal database project (RDP) Classifier algorithm against the Silva 128/16s bacterial database using threshold for confidence of 70%.

### 2.6. Statistical Analysis 

The data of the reproductive and growth performance, antioxidant and inflammation were analyzed using the general linear model (SPSS v. 22.0 for Windows; SPSS Inc., Chicago, IL, USA). One sow in a pen was considered as the experimental unit of analyses for the difference in reproductive performance. As for inflammatory cytokines, antioxidants and microbial analysis, individual pigs were considered as the experimental unit. The difference in alpha diversity was tested using Kruskal-Wallis test (SPSS v. 22.0 for Windows; SPSS Inc., Chicago, IL, USA) and *p*-values were adjusted with FDR (below 5%) [19]. *p*-values below 0.05 were considered statistically significant and all data were presented as mean ± SEM. Beta-diversities based on the Bray-Curtis and non-metric multidimensional scaling (NMDS) were calculated. Linear discriminant analysis (LDA) effect size (LEfSe) analysis was used to identify the differential genera (only genera with an average relative abundance greater than 0.03% were considered). And the multi-group comparison strategy was one against-all. Correlations between bacterial communities and plasma parameters were assessed by Spearman’s correlation analysis using the “heatmap” and data were expressed as mean values.

## 3. Results

### 3.1. Effect of REE Supplementation on Reproductive Performances of the Sows and Growth Performances of Their Piglets

As displayed in Table 2, there were no differences in total litter size, the number of piglet born alive, average birth weight and total piglet birth weight between the two groups. But the REE-G group showed a significantly reduction in CV of within-litter birth weight (*p* < 0.01), indicating an improvement of uniformity with the intervention of REE mixture during late gestation.

The results in Table 3 showed that the average weaning weight in REE-L+ and REE-L- group on day 21 of lactation were significant higher (*p* < 0.01) than that of the control group. Since there was no difference in birth weight, the difference in weaning weight could attribute to the higher average daily gain (ADG) in REE-L+ and REE-L- group than that of the control group (*p* < 0.01). There was no difference in growth hormone (GH) secretion among piglets from the three groups (Figure 1). The plasma insulin-like growth factor-1 (IGF-1) level in the REE-L+ group was significantly higher (*p* < 0.01) than that in the control group, while no difference between the REE-L- and the control group.

### 3.2. Effect of REE Supplementation on Antioxidant Capacity of the Sows and Their Offspring

The antioxidant capacities in plasma with REE supplementation were presented in Figure 2. For the farrowing sows, there was an increase in activity of T-SOD (*p* < 0.05), CAT (*p* < 0.01), and content of GSH-Px (*p* < 0.01) in response to the REE supplementation during late gestation. T-AOC showed non-significant changes between the two groups. MDA showed a downtrend (*p* = 0.082) in the REE-G group compared with the control group. For the lactating sows, the plasma content of T-AOC, GSH-Px, CAT in REE-L+ group (*p* < 0.01), and REE-L- (*p* < 0.05) group were significantly higher than that of the control group, but there was no difference between the REE-L+ and REE-L- groups. For weaning piglets, no significant difference was observed in plasma T-AOC, GSH-Px, CAT, and MDA content on day 21 of lactation among the three groups. Only plasma content of T-SOD in piglets of the REE-L+ group was higher (*p* < 0.05) than that of the control group, but there was no difference between the REE-L- group and control group. The results indicated a limited enhancing effect of maternal REE supply on antioxidant capacity of the piglets. Besides, further addition of REE was recommended during lactation.

### 3.3. Effect of REE Supplementation on Inflammation-Related Indicators in the Sows and Their Offspring

As demonstrated in Figure 3, REE supplementation during late gestation had no impact on plasma levels of IL-10, IL-1β, and TNF-α of the farrowing sows. For lactating sows, there was no significant change in level of IL-10 and IL-1β among the three groups. However, the level of TNF-α in sows of the REE-L+ group decreased significantly (*p* < 0.01), in comparison to the control group. For weaning piglets, plasma IL-10 and IL-1β levels did not show differences among the three groups. However, the plasma level of pro-inflammatory factor TNF-α was greatly reduced in piglets from the REE-L+ and REE-L- group (*p* < 0.01; Figure 3) compared with the control group, while there was no difference between the REE-L- and REE-L+ group.

### 3.4. Effect of REE Supplementation on Fecal Microbiota of the Sows and Their Offspring

For lactating sows, a total of 897,910 sequences were generated from 24 fecal samples after noise sequences clearance. Based on 97% sequence similarity, 912 operational taxonomic units (OTUs) were identified and then assigned to 17 phyla, 34 classes, 60 orders, 92 families, 216 genera, and 361 species. For piglets, a total of 1,094,555 sequences were generated from 24 fecal samples, 925 OTUs were identified and then assigned to 23 phyla, 42 classes, 71 orders, 120 families, 311 genera, and 548 species.

The differences in alpha-diversity of fecal microbiota in lactating sows and their piglets among the three groups were shown in Table 4. For both of the sows and the piglets, REE supplementation did not alter the fecal microbial diversity (Simpson, Shannon index) and richness (Sobs, ACE index). For beta-diversity analysis of sows, an NMDS plot based on Bray-Curtis distance (Appendix A) revealed the REE-L+ group sows had a distinct microbiota composition from that of the control sows. Meanwhile, REE-L+ group separated from the REE-L- group sows. However, REE-L- group could not be distinguished from the control group. As well, the microbiota of piglets from the REE-L+ group was separated from those in the control group (Appendix A).

At the phylum level, the relative abundance of *Firmicutes* in lactating sows was higher in REE-L- group (81.0%) than the control group (78.2%), and *Bacteroidetes* was higher in REE-L+ group (19.1%) than he control group (13.0%) (Appendix A). In piglets, *Firmicutes* and *Bacteroidetes* were the dominant phyla in all three groups. And the abundance of *Firmicutes* phylum were higher, while *Proteobacteria* phylum were lower in REE-L- (5.3%) and REE-L+ group (6.7%), compared to the control group (14.8%) (Appendix A). 

Abundant genera of fecal microbiota in (Figure 4A) sows and (Figure 4B) their piglets. Significant differences in relative abundance of the fecal microbiota from phyla to genera were further identified using the LEfSe analysis. In lactating sows, proportions from the *Spirochaetae* phylum to *Spirochaetaceae* family were increased in REE-L+ group compared with the control group. *Treponema_2*, *Christensenellaceae_R-7_group*, *Prevotellaceae_UCG-001*, *Ruminiclostridium_1*, *Turicibacter*, *norank_f__Bacteroidales_BS11_gut_group* genus was enriched in REE-L+ (Figure 5A), while *Clostridia* class from *Firmicutes* phylum and *Lachnospiraceae_XPB1014_group* genus showed an increased abundance in the control group. On the other hand, REE-L- group had little difference with the control (Figure 5B). Besides, the abundance of *Succinivibrio*, *Desulfovibrio*, *Phascolarctobacterium*, *Prevotella_1* genus, and *Rikenellaceae* family in REE-L+ group were higher than that of REE-L- group (Figure 5C).

In piglets, Epsiloproteobacteria family and the genus Campylobacter, Micrococcus, Prevotella_9, Tyzzerella was observed to be enriched in the control group. Ruminococcaceae_UCG-005, Ruminococcaceae_UCG-002, Ruminococcaceae_UCG-014, Lachnospiraceae_FCS020_group, and Anaerotruncus genus were significantly higher in REE-L+ group than in the control group (Figure 5D). More abundant genera Campylobacter, Helicobacter, Hungatella, Eisenbergiella and family Rikenellaceae and phylum Proteobacteria were enriched in the control group when compared with REE-L- group (Figure 5E).

### 3.5. Correlation of Gut Microbiota with Plasma Parameters of Sows and Their Piglets

As shown in Figure 6A, for the lactating sows, the Spearman’s correlation matrix illustrated that the genus *Lachnospiraceae_XPB1014_group* was negatively correlated with the T-AOC (R = −0.497, *p* < 0.05). The genus *Turicibacter* was positively associated with CAT (R = 0.632, *p* <0.01) and IL-10 (R = 0.423, *p* < 0.05). The genus *Treponema_2* was positively correlated with T-AOC (R = 0.41, *p* < 0.05). For piglets (Figure 6B), the correlation analysis revealed that the genus *Campylobacter* was negatively correlated with T-AOC (R = −0.491, *p* < 0.05), T-SOD (R = −0.415, *p* < 0.05) and positively correlated with MDA (R = 0.448, *p* < 0.05), TNF-α (R = 0.497, *p* < 0.05), and IL-1β (R = 0.533, *p* < 0.01). The genus *Hungatella* was negatively associated with T-AOC (R = −0.422, *p* < 0.01), and the genus *Ruminococcaceae_UCG-005* was negatively correlated with IL-10 (R = −0.405, *p* < 0.05), TNF-α (R = −0.611, *p* < 0.01), and IL-1β (R = −0.511, *p* < 0.05). The genus *Ruminococcaceae_NK4A214_group* showed positive relation to T-AOC (R = 0.533, *p* < 0.01), CAT (R = 0.538, *p* < 0.01) and GH (R = 0.466, *p* < 0.05). The genus *Anaerotruncus* and *Turicibacter* was positively and negatively correlated with GH, respectively (R = 0.493 vs. −0.425, *p* < 0.05). *Ruminococcaceae_UCG-002* was negatively related with MDA (R = −0.423, *p* < 0.05) but positively with T-SOD (R = 0.428, *p* < 0.05) and ADG (R = 0.555, *p* < 0.01).

## 4. Discussion

It was reported that REEs have already been widely applied in livestock [20], however, in the case of pigs, most of the research was carried out on weaned, growing, and fattening pigs [8,11], with no reports about effects of REEs on sows and their offspring. In the present study, we investigated the effects of maternal REE supply during late gestation and lactation on performances and health of the sows and their nursing piglets, the study also looked at the impact on fecal microbiota of the sows and piglets. Besides, we observed an improved uniformity of birth weight at delivery and ADG of piglets during the suckling stage, an elevated plasma antioxidase activity, and a decreased TNF-α content in sows and piglets. Apart from the improvement in the health and performances, we also observed an alteration in composition of the fecal microbiota, the intestinal probiotic colonization in lactating sows and weaning piglets were increased. To the best of our knowledge, this study is the first to report that REE supplementation during the perinatal stage has beneficial roles on both the sows and their offspring.

A previous study indicated that dietary Ce supplementation in female rabbits [21] increase the total litter weight at birth, and post-partum weight of the offspring. In vitro research on mice also proposes an increased proportion of the developed embryo cells under the intervention of La [22]. The present study demonstrated that REE supplementation on sows during late gestation reduced the CV of within-litter birth weight, indicating an improvement of reproductive performance. Potential interaction of dietary REEs with reproductive hormones and vasodilation of placental vessels were suspected as the reasons for the improved fertility observed in our study. In addition, maternal REE supply significantly boosted the growth of their nursing piglets, also our results indicated a trans-generationally beneficial effect for REE supply during late gestation on the growth of their offspring during lactation, no depending on whether the REE was continuously supplied during the lactation or not. In mammals, postnatal growth is controlled by the activity of the somatotropic axis, where GH instructs the liver and peripheral tissues to produce IGF-1 to promote growth [23,24], IGF-1 not only mediates the growth-promoting effects of GH, but also promotes anabolism [25]. An earlier study showed that the level of GHs in serum and ADG are increased when incorporating La to barrow’s diet [9]. Our results confirmed the growth-promoting effect of REEs on their offspring and improvement of IGF-1 in plasma but no difference in GHs was observed. The similar beneficial role on ADG but different results on level of GH between these two studies might be due to the complexity of GH/IGF-I system with growth regulation at multiple levels [26], and the different animal breeds. In our study, improvements observed in the ADG of the piglets can be attributed to the REEs stimulate secretion of IGF-1, which is considered as an important growth factor. Furthermore, our results showed that the level of IGF-1 in the REE-L+ group was significantly higher while no difference between the REE-L- and control group, indicating a specific beneficial role of further REE supplementation during lactation. 

In production, oxidative stress can give rise to suboptimal livestock health conditions and economic loss. Especially, oxidative status of the sows is not only closely related to their own health, but also that of their nursing piglets [27,28]. The antioxidant effects of REE have been previously reported and confirmed [29,30,31,32,33] on various kinds of animals. In the current study, REE supply elevated the antioxidant enzyme activity of the gestating and lactating sows indicated by higher plasma CAT, GSH-Px, etc., moreover, our results suggested a continuously beneficial effect of REE during late gestation on subsequent antioxidant capacity of lactating sows. However, only T-SOD content of piglets from REE-L+ group differed considerably from the control, indicating a limited improvement on antioxidant capacity of the piglets through maternal REE supply during late gestation and lactation.

Cytokines play a vital role in the immune and inflammatory responses and therefore their balance is crucial for the health and protection against infection [34]. Studies in pigs have suggested that the immune system is active in pregnant sows [35] and neonatal piglets [36]. In this study, we found no differences in all these cytokines between the REE-G and the control group, with only the level of TNF-α in sows from the REE-L+ group significantly decreased compared to the control while no difference between the REE-L- and the control group was found. This indicated the intergenerational advantageous effect of REE supplementation to a sow’s diet during late gestation on immune response of their piglets, and among the cytokine we measured, TNF-α was the only one responsive to REE. TNF-α plays a central role in pathogenesis of a broad range of inflammatory diseases [37]. An earlier research on mice reported that LaCl_3_ is a potent inhibitor of pro-inflammatory factors and greatly decreases the secretion of TNF-α and IL-1β [15]. The present study suggested that maternal REE supply during late gestation and lactation decreased the level of plasma TNF-α in both sows and the piglets yet no difference in IL-1β, which confirmed the effects of REEs on suppressing excessive immune response in pigs. Based on our results, further addition of REE during lactation helped to maintain a better antioxidant and immune status for both the lactating sows and their offspring.

The role of intestinal microbiota in host physiological health has been recognized [38,39], and pioneers in gut microbiology have emphasized the significance of diet: the microbe interactions may contribute to health status [40]. In this study, *Clostridia* were abundant in sows from the control group, and Clostridial flagella were known to exert a peculiar role in adhesion and pathogenicity [41]. In addition, the higher abundance of *Lachnospiraceae_XPB1014_group* found in the control group was shown negatively related with T-AOC. Besides, this bacteria is reported in a previous study to be associated with the gut dysfunction in humans and mice [42]. As for the bacteria enriched in sows from the REE-L+ group, *Christensenellaceae_R_7_group* is regarded as a potentially beneficial bacteria due to its positive role in intestinal environment and immunomodulation [43]. *Turicibacter* was positively correlated with IL-10 and CAT in our study. It can degrade polysaccharides to short chain fatty acids (SCFAs) such as butyrate. Changes in these “butyrogenic” bacteria [44] may in turn influence metabolic, inflammatory bowel disorders. Indeed, butyrate, which can be used as a source of energy by the host, confers many benefits including anti-inflammatory effects [45]. Likewise, *Phascolarctobacterium* enriched in REE-L+ is a Gram-negative genus able to produce SCFAs.

Environmental factors and maternal bacteria quickly colonize offspring gut after parturition [46] through the birth canal, breast feeding, or skin contact [47] and shape the onset of an intestinal immune system and its future development. Remarkably, REE supplementation reduced the presence of *Proteobacteria* in offspring weaned piglets, which can be considered advantageous because the prevalence of *Proteobacteria* always indicates a labile microbial community and is associated with intestinal inflammation [48]. This phylum includes a variety of bacteria known to cause intestinal pathology in humans and animals [49]. Similarly, *Campylobacter* spp., some of which associated with diarrhea in piglets [50], were shown to be more abundant in the control group, and positively related to TNF-α but negatively related to T-SOD and T-AOC. Fortunately, opportunistic pathogens such as *Proteobacteria*, *Campylobacter* were suppressed in the REE supplied group. *Ruminococcaceae_UCG-005*, *Ruminococcaceae_NK4A214_group*, and *Ruminococcaceae_UCG-002* showing the positive relation with antioxidase and the negative relation with pro-inflammatory factor, were increased in piglets of the REE-L+ group. Besides, *Ruminococcaceae_UCG-002* genus was significantly associated to ADG of the suckling piglets, which is consistent with a recent study reporting that the genus is positively correlated with the body weight of newborn piglets [51]. In a recent report assessing gut bacterial composition in suckling piglets, the most discriminant bacterial family is the high abundance *Ruminococcaceae* in feces of healthy pigs when compared with the diarrhea pigs [52], which is in line with our study. Our results showed that *Ruminococcus* (a SCFAs-producing bacterium) was found to be abundant in the REE-L+ group. As mentioned before, SCFAs can provide most of the energy needed by colonic epithelial and gut immune cell, and also play roles in anti-inflammation, antioxidant, and mucosal protection [53]. These further support the idea that beneficial alterations in the gut microbiome lead to changes in oxidative stress and inflammatory markers that affect performance and health. In fact, the antibacterial activities of rare earth ions have been previously reported [54], a possible explanation could be that lanthanides impede the physiological activities of undesirable bacterial species by strongly inhibiting the bacterial respiration and destroy the micro-structure of cell membrane [14,55].

Our results ultimately suggest that maternal REE supplementation during late gestation and lactation may regulate oxidative stress and immune status by modulating gut bacteria and inhibiting the proliferation of potential pathogen in sows and accordingly modulate the microbiota establishment in their offspring, thus promoting hormone secretion and improving the health condition of suckling piglets. It is informed that La^3+^ or Ce^3+^ has ionic radii similar to important nutritional ion Ca^2+^, and specifically regulate the activities of various biochemical pathways and affect intermediate metabolism [13]. That may be one of the reasonable explanation for the results of our study. Anti-bacteria, anti-oxidative, and anti-inflammatory properties ascribed to REEs included in sows’ diets may also contribute to performance enhancing effects of offspring piglets. Whereas, the exact mechanisms underlying these beneficial effects of REEs on pigs, just as how the REEs included in diets of the sows influence piglets’ biological responses, remain to be verified and further research is required. Even so, the results obtained from our study may be supported by a recent review, reporting the lactocrine hypothesis for maternal programming of reproductive development in pigs. The hypothesis provided a lactocrine mechanism, by extending trophic signals from mother into offspring in colostrum [56]. Apart from that, whether sows or piglets, little difference was found between the REE-L- and the control group in the current study, indicating the necessity of further addition of REEs during the lactation stage.

## 5. Conclusions

Taken together, REE supplementation improves the antioxidant, immune status of sows and beneficially changes microbial composition. We speculate these lead to a good physiological condition of sows and, accordingly, improve the reproductive performance of sows and growth performance of their offspring. The increased ADG of piglets is presumably resultant from the rising secretion of IGF-1 and inhibition the potential pathogens by maternal REE addition. Furtehrmore, it is more effective to add an REE mixture during both late gestation and lactation rather than only in late gestation. The findings of our study offer new insights into how an REE diet during late gestation and lactation profoundly affects porcine health and performance, and also will facilitate the application of REEs in animal production.

## Figures and Tables

**Figure 1 animals-09-00738-f001:**
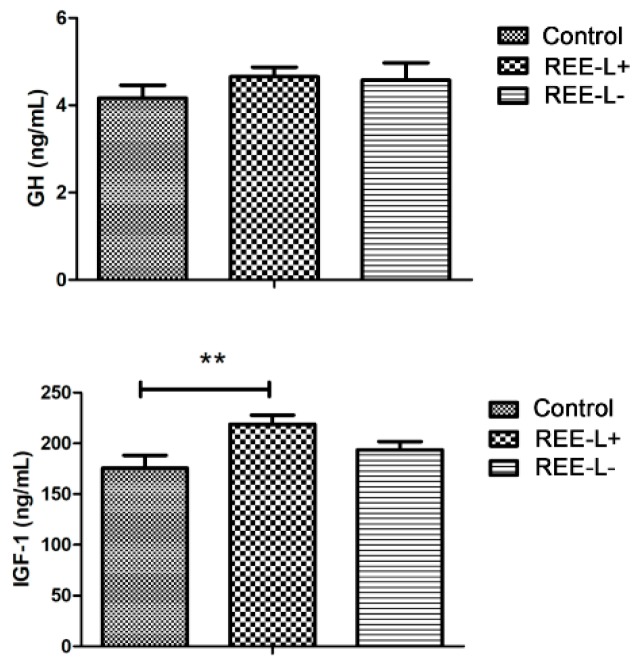
Effect of maternal REE supplementation on growth hormone (GH) and insulin-like growth factor-1 (IGF-1) secretion of their piglets. *n* = 8 per group. Values are mean ± SEM. ** *p* < 0.01. REE-L+, REE supplied during gestation and lactation. REE-L-, REE supplied only in late gestation.

**Figure 2 animals-09-00738-f002:**
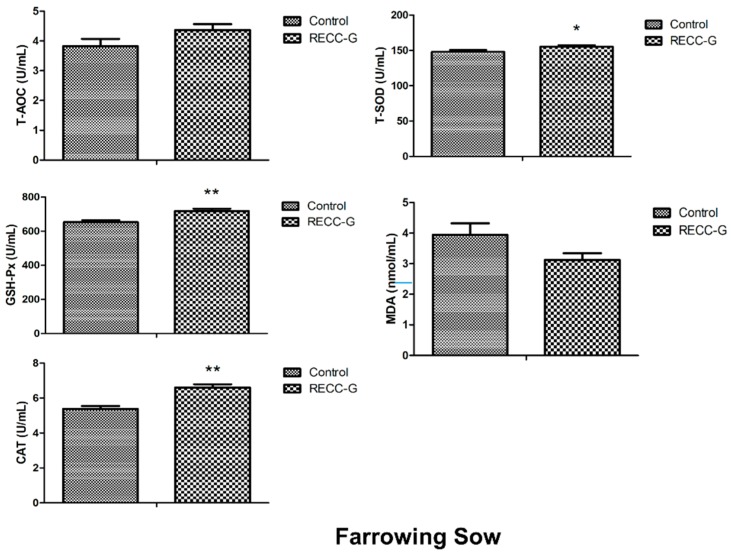
Effect of maternal REE supplementation on plasma antioxidant defense system of the sows and their piglets. *n* = 8 per group. Values are mean ± SEM. * *p* < 0.05, ** *p* < 0.01. REE-L+, REE supplied during gestation and lactation. REE-L-, REE supplied only in late gestation.

**Figure 3 animals-09-00738-f003:**
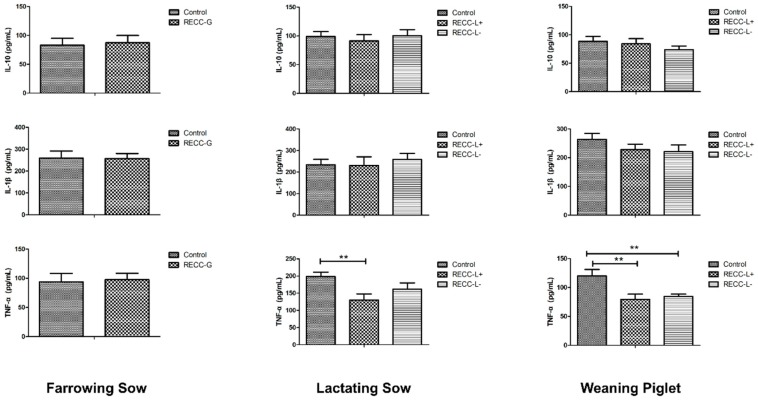
Effect of maternal REE supplementation on plasma inflammatory cytokines of the sows and their piglets. *n* = 8 per group. Values are mean ± SEM. ** *p* < 0.01. REE-L+, REE supplied during gestation and lactation. REE-L-, REE supplied only in late gestation.

**Figure 4 animals-09-00738-f004:**
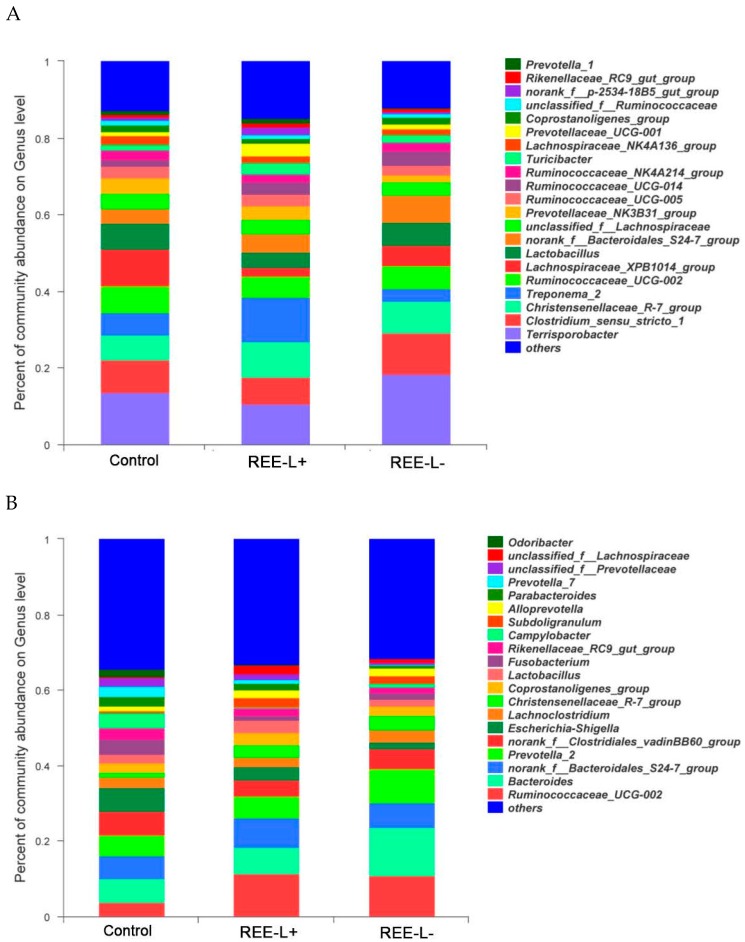
Abundant genera of fecal microbiota in (**A**) sows and (**B**) their piglets. *n* = 8 per group. REE-L+, REE supplied during gestation and lactation. REE-L-, REE supplied only in late gestation.

**Figure 5 animals-09-00738-f005:**
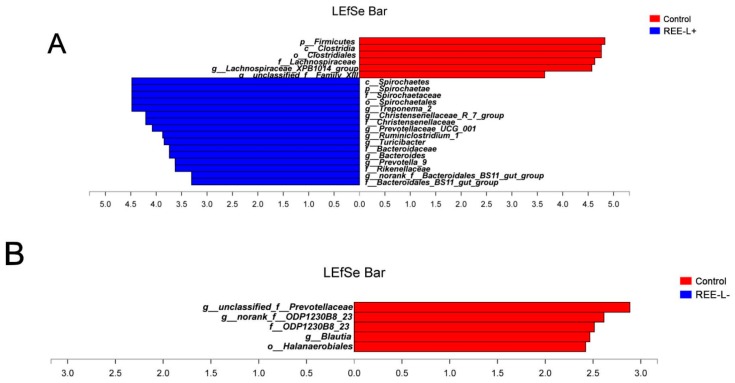
Linear discriminant analysis effect size (LEfSe) analysis of the different abundance of microbiota in fecal samples of (**A**–**C**) sows and (**D**–**F**) their piglets from phylum to genus levels. Histograms of a linear discriminant analysis (LDA) score (threshold > 2) are plotted. *n* = 8 per group. REE-L+, REE supplied during gestation and lactation. REE-L-, REE supplied in late gestation.

**Figure 6 animals-09-00738-f006:**
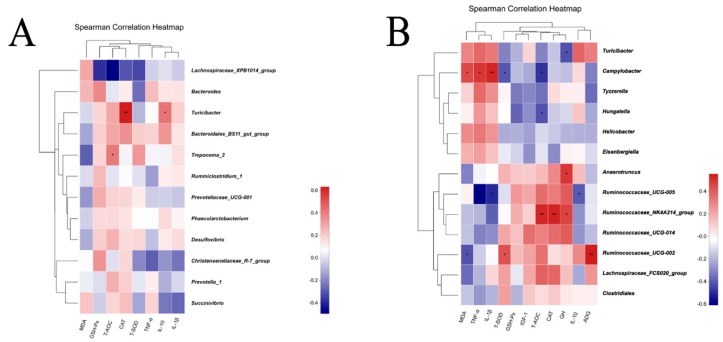
Spearman correlation analysis between differential genera and oxidative stress, inflammatory marker and growth performance of (**A**) the sows and (**B**) their piglets. *n* = 8 per group.

**Table 1 animals-09-00738-t001:** Composition of diet for sows during the late gestation and lactation.

Ingredients	Late Gestation	Lactation
Corn, %	-	46.54
Red sorghum, %	25.00	25.00
Unprocessed barley, %	46.72	-
Rice bran meal, %	10.00	-
Soybean meal, %	9.28	22.31
Palm meal, %	5.00	-
Soymeal oil, %	-	1.58
Mountain flour, %	1.48	1.60
Monocalcium phosphate, %	0.83	1.09
Sodium chloride, %	0.47	0.51
Sodium bicarbonate, %	0.30	0.30
Lysine, %	0.20	0.34
DL-Methionine, %	0.08	0.07
L-Threonine, %	0.10	0.10
L-Tryptophan, %	-	0.02
Choline chloride, %	0.14	0.14
Premix ^a^, %	0.40	0.40
Nutrient composition		
Dry matter, %	89.02	87.51
Crude protein, %	13.20	16.50
Ether extract, %	2.18	4.09
Crude ash, %	6.13	5.49
Crude fiber, %	4.92	2.53
Calcium, %	0.77	0.85
Total phosphate, %	0.69	0.54
Available phosphate, %	0.30	0.34
(Standard Total Tract Difestibility) STTD phosphate, %	0.32	0.33
Net energy, kJ/kg	2178.20	2511.40

^a^ The premix provided the following per kg of diets: Cu, 250 mg; Fe, 150 mg; Zn, 200 mg; Mn, 40 mg; Se, 0.4 mg; I, 0.3 mg; vitamin E, 20 mg; vitamin A, 11250 IU; vitamin D_3_, 2500 IU; vitamin K_3_, 2.5 mg; vitamin B_1_, 2.5 mg; vitamin B_2_, 6.0 mg; vitamin B_6_, 3.0 mg; vitamin B_12_, 0.08 mg; biotin, 0.01 mg; pantothenic acid, 12.5 mg; folic acid, 1.25 mg; niacin, 25 mg.

**Table 2 animals-09-00738-t002:** Effects of rare earth elements (REE) supplementation during late gestation on reproductive performances of the sows at birth.

Items	Dietary Treatment		*p*-Value
	Control	REE-G ^a^	
Total litter size, *n*	13.43 ± 0.35	13.21 ± 0.39	0.686
Born alive, *n*	13.20 ± 0.35	12.92 ± 0.38	0.593
Average birth weight of piglets/litter, kg	1.44 ± 0.03	1.51 ± 0.03	0.093
Total birth weight/litter, kg	19.29 ± 0.41	19.49 ± 0.52	0.768
Within-litter birth weight CV ^b^, %	0.21 ± 0.01	0.18 ± 0.01 **	<0.01

** *p* < 0.01 versus the control group. ^a^ REE-G, REE supplied during gestation. ^b^ CV, Coefficient of variation.

**Table 3 animals-09-00738-t003:** Effects of REE supplementation during the late gestation and lactation on growth performances of the piglets.

Items	Dietary Treatment			*p*-Value
	Control	REE-L+ ^a^	REE-L- ^b^	
Weight at 21st day, kg	5.71 ± 0.12	6.21 ** ± 0.14	6.26 ** ± 0.12	<0.01
Daily weight gain, g/day	223.06 ± 4.88	241.75 ** ± 5.84	240.07 ** ± 5.70	<0.01

** *p* < 0.01 versus the control group. ^a^ REE-L+, REE supplied during both late gestation and lactation. ^b^ REE-L-, REE supplied only in late gestation.

**Table 4 animals-09-00738-t004:** Alpha-diversity of fecal microbiota from the lactating sows and the weaning piglets on day 21 of lactation.

Sample	Control	REE-L- ^a^	REE-L+ ^b^	*p*-Value
Sow				
Sobs	494.63 ± 10.88	498.25 ± 13.36	519.13 ± 9.02	0.22
Ace	592.28 ± 13.10	611.76 ± 8.56	613.87 ± 7.13	0.34
Shannon	4.06 ± 0.11	3.89 ± 0.17	4.28 ± 0.08	0.15
Simpson	0.06 ± 0.01	0.08 ± 0.02	0.04 ± 0.01	0.14
Piglet				
Sobs	382.50 ± 32.54	294.88 ± 20.84	359.13 ± 15.12	0.05
Ace	466.70 ± 37.06	371.10 ± 29.15	437.04 ± 17.96	0.08
Shannon	3.91 ± 0.10	3.51 ± 0.18	3.88 ± 0.09	0.23
Simpson	0.05 ± 0.01	0.09 ± 0.02	0.05 ± 0.01	0.11

^a^ REE-L+, REE mixture supplied during both late gestation and lactation. ^b^ REE-L-, REE mixture supplied only in late gestation.

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
