# Peer review of "Effects of Maternal Supplementation with Rare Earth Elements during Late Gestation and Lactation on Performances, Health, and Fecal Microbiota of the Sows and Their Offspring"

_animals, 2019, doi:10.3390/ani9100738_

Round 1
Reviewer 1 Report
There are many times when past tense words are used and it should be present tense, the most common is using the word "were" (past tense) when it should be "are" (present tense).
You did not use corn-soybean meal-based diets as stated in the materials and methods. There is no corn in the gestation diet. Also, there is more sorghum in the lactation than soybean meal. So, your diets are "grain-based" or clarify the gestation is Barley/Sorghum and the lactation is Corn/Sorghum and Soybean Meal
Why two programs to run statistics? SPSS and SAS can do the same analysis? Not sure why you run general linear models in one and your Kruskal-Wallis in another. Also need to supply the company contact information in parenthesis for these two programs. Your statistical analysis methods need clarification.
Also, you need to capitalize Bray-Curtis throughout the paper.
In your results section, do not repeat the methods. The first line in almost each section tells us what you did, but that information has already been given.
Good graphs and figures.
Your conclusion needs some editing to be more concise.
Author Response
Point 1: There are many times when past tense words are used and it should be present tense, the most common is using the word "were" (past tense) when it should be "are" (present tense).
Response 1: We have carefully revised the manuscript according to your comment on tense.
Point 2: You did not use corn-soybean meal-based diets as stated in the materials and methods. There is no corn in the gestation diet. Also, there is more sorghum in the lactation than soybean meal. So, your diets are "grain-based" or clarify the gestation is Barley/Sorghum and the lactation is Corn/Sorghum and Soybean Meal.
Response 2: Thank you very much for pointing out our misrepresentation about diets in the Materials and Methods, and we have revised it to “grain-based diet” according to your considerate suggestion (page 3).
Point 3: Why two programs to run statistics? SPSS and SAS can do the same analysis? Not sure why you run general linear models in one and your Kruskal-Wallis in another. Also need to supply the company contact information in parenthesis for these two programs. Your statistical analysis methods need clarification.
Response 3: After inspection, we found that only the data of alpha diversity were run by SAS software. To unify the statistical analysis methods, we ran the data of alpha diversity using SPSS (SPSS v. 22.0 for Windows; SPSS Inc., Chicago, IL, USA) with Kruskal-Wallis test, the results showed the same significance. Also in the paper, the statistical analysis methods have been further clarified and highlighted in red according to the reviewer’s comments (page 5).
Point 4: Also, you need to capitalize Bray-Curtis throughout the paper.
Response 4: We have capitalized Bray-Curtis throughout the article.
Point 5: In your results section, do not repeat the methods. The first line in almost each section tells us what you did, but that information has already been given.
Response 5: As suggested, the repeated statements of each paragraph in the results section were deleted (page 5-11).
Point 6: Good graphs and figures.
Response 6: Thank you very much for your recognition.
Point 7: Your conclusion needs some editing to be more concise.
Response 7: We have edited the conclusions to present a more concise text according to the reviewer's comment, and marked as red in the revised manuscript. (page 14).
Reviewer 2 Report
I think the use of RARE earth elements in pig feed has no production or practical significance.
Author Response
Point 1: I think the use of rare earth elements in pig feed has no production or practical significance.
Response 1: The introduction part of our original manuscript might not be clear enough about the production or practical significance of REE supplementation. In view of this, we have strengthened the introduction to highlight the physiological and nutritional importance of the use of these products in pig feed.
As a matter of fact, the application of REE as feed additive in pig production has been practiced for a long time. There are many literatures concerning the performance enhancing effects of REE for pigs and many more have been reviewed by Redling, Rambeck and Wehr [1]. In the research reports, body weight gain was shown to be improved by 5 to 23% and feed conversion between 4 and 19% under the influence of REE, significant improvements in pig performance were observed [2,3]. Additionally, physiological properties of REE including enhanced enzyme and hormone activity, improved metabolism and oxidative status, anti-inflammatory and anti-bacteria actions have also been proposed [4]. Moreover, the innovation of this study was to observe the biological responses of sows and their offspring under the intervention of maternal REE supply. This is different from previous researches, which have focused on exploring the growth-promoting effects of REE on growing or finishing pigs [5,3]. Our results also show an extensively beneficial effects including beneficial changes in the gut microbiota as well as a series of positive changes in plasma index (antioxidase, inflammatory cytokines, IGF-1) of sows and their piglets, and improving the litter uniformity. Furthermore, this study gives a new insight to improve the reproductive performance of the perinatal sows. Therefore, the use of REE as feed additives to maintain swine health and performance has certain industrial significance.
References:
[1] Rambeck WA, Wehr U: Rare earth elements as alternative growth promoters in pig production. Arch Tierernahr 2000, 53:323–334.
[2] He ML, Rambeck WA: Rare earth elements: a new generation of growth promoters for pigs. Arch Anim Nutr 2000, 53:323–334.
[3] He ML, Ranz D, Rambeck WA: Study on the performance enhancing effect of rare earth elements in growing and finishing pigs. J Anim Physiol Anim Nutr 2001, 85:263–270.
[4] Redling K: Rare earth elements in agriculture with emphasis on animal husbandry. Muenchen: Diss Ludwig-Maximilians-Universitaet; 2006:325.
[5] Cai L, Nyachoti C M, Kim I H. Impact of rare earth element-enriched yeast on growth performance, nutrient digestibility, blood profile, and fecal microflora in finishing pigs. Canadian Journal of Animal Science, 2018, 98.
Reviewer 3 Report
This paper was conducted to evaluate the effect of maternal supplementation with REE on sows and piglets productivity, antioxidant capacity, immunity and gut microbiota. Even though the article provides interesting information, specially that related with antioxidant capacity, immunity and gut microbiota of sows and the progeny, there are some key issues that need to be revised by the authors as they may be influencing some of the results reported.
Firstly, the section "simple summary" (mandatory for papers in Animals) is missing. By itself this should be a reason for paper rejection. However, probably the most important issue is about experimental design. It is not clear to me why the authors performed the experiments in that way with a control group (n=60), REE-L(-) group (n=30) and REE-L(+) group (n=30) during lactation, and why they did not design a 2x2 classical factorial design, with 2 levels of REE inclusion during gestation, and 2 leveles of REE inclusion during lactaction. This would have allowed a more comprehensive and equally distributed study. From my perspective, this study follows a design with 3 different treatments: control, REE during gestation only, and REE during gestation and lactation. Nevertheless, they are not comparable.
In the same way, it is informed that sows age ranged from 3-5 parturition, and it is not clear whether the authors considered that factor for the distribution within groups. Sows age (or number of parturitions) may influence their productivity in terms of piglets born, birth weight and weaning weight. Therefore, if this factor was not considered and/or are not equally distributed, all productive results might be affected by that noise.
In that case, I would consider to continue with the paper just considering the results coming from "samples", that might sound well obtained to me (8 samples per treatment).
Apart from that, there are different suggestions within the manuscript that need to be addressed before continuing with publication. One critical is about the physiological (or nutritional) basis of this work. How do the REE included in the diets of sows influence piglets' biological responses? I am missing much information about this in the Discussion.
If the authors can address this previous comments I would be happy to reconsider my decision.

Author Response
Point 1: This paper was conducted to evaluate the effect of maternal supplementation with REE on sows and piglets productivity, antioxidant capacity, immunity and gut microbiota. Even though the article provides interesting information, specially that related with antioxidant capacity, immunity and gut microbiota of sows and the progeny, there are some key issues that need to be revised by the authors as they may be influencing some of the results reported.
Response 1: Thanks for your positive comments.
Point 2: Firstly, the section "simple summary" (mandatory for papers in Animals) is missing. By itself this should be a reason for paper rejection. However, probably the most important issue is about experimental design. It is not clear to me why the authors performed the experiments in that way with a control group (n=60), REE-L(-) group (n=30) and REE-L(+) group (n=30) during lactation, and why they did not design a 2x2 classical factorial design, with 2 levels of REE inclusion during gestation, and 2 levels of REE inclusion during lactation. This would have allowed a more comprehensive and equally distributed study. From my perspective, this study follows a design with 3 different treatments: control, REE during gestation only, and REE during gestation and lactation. Nevertheless, they are not comparable.
Response 2: With respect to the section "simple summary", we have actually attached it in the original manuscript, but we are not sure about the specific reasons for its omission in the paper.
In view of this, we’ve re-added “simple summary” and marked it in red.
Regarding the experimental design, 2x2 classical factorial design is good, and the experimental design from the perspective of statistical needs should indeed be done to identify the effects of REE during gestation and lactation. Nevertheless, for other purpose, we didn't do it. It’s informed that, during late gestation, maternal glucose and free fatty acid concentrations increase, allowing for greater substrate availability for fetal growth [1]. At the same time, possible nutritional interventions are effective at increasing uniformity in fetal weights and regulating gut microbial to prevent sow constipation during late pregnancy [2,3]. The late gestation shows more important physiological significance. Basically, our study first or mainly look at the positive effects of REE supplementation during late pregnancy, therefore, with the exception of the control group, REE was added to the other two groups during late gestation. As for the REE-L+ group (add both in late gestation and lactation), the point is to see if it is necessary to continue to add REE during lactation by comparing with those added only in late gestation. In the case of adding REE only during lactation, we believe that this kind of exploration is not too meaningful for the industry from the perspective of pig production, so there is no such group. Expecting such a response, the referees can be satisfied.
References:
[1] Lain K Y, Catalano P M. Metabolic Changes in Pregnancy. Clinical Obstetrics and Gynecology, 2007, 50(4):938-948.
[2] Wang J, Feng C, Liu T, et al. Physiological alterations associated with intrauterine growth restriction in fetal pigs: Causes and insights for nutritional optimization. Molecular Reproduction and Development, 2017.
[3] Zhang Y, Lu T, Han L, et al. L-Glutamine Supplementation Alleviates Constipation during Late Gestation of Mini Sows by Modifying the Microbiota Composition in Feces. BioMed Research International, 2017, (2017-3-12), 2017, 2017(1):4862861.
Point 3: In the same way, it is informed that sows age ranged from 3-5 parturition, and it is not clear whether the authors considered that factor for the distribution within groups. Sows age (or number of parturitions) may influence their productivity in terms of piglets born, birth weight and weaning weight. Therefore, if this factor was not considered and/or are not equally distributed, all productive results might be affected by that noise. In that case, I would consider to continue with the paper just considering the results coming from "samples", that might sound well obtained to me (8 samples per treatment). How was the sows age factor equally distributed with your experimental design?
Response 3: Certainly, this factor was carefully considered in our study. The reason why we select parity 3-5 sows is that 1st or 2nd parity is relatively special with a poor reproductive performance. On the whole, the litter variation of parity 3-5 sows is little with a high litter size [1]. Among the 3 groups, sows of different parity were equally distributed in each group. For example, in the control group, sows of parity 3,4 and 5 each account for one-third of the number of sows, and the other two groups are in the same way. The same is true for the distribution of each parity in the final sampled sows. Overall, the parity of sow was kept consistent within the same group and among the different groups.
Reference:
[1] Hoving L L, Soede N M, Graat E A M, et al. Reproductive performance of second parity sows: Relations with subsequent reproduction. Livestock Science, 2011, 140(1-3):124-130.
Point 4: Apart from that, there are different suggestions within the manuscript that need to be addressed before continuing with publication. One critical is about the physiological (or nutritional) basis of this work. How do the REE included in the diets of sows influence piglets' biological responses? I am missing much information about this in the Discussion.
Response 4: Our results indicate that maternal REE supplementation during late gestation and lactation improve the oxidative and immune status by modulating gut bacteria and inhibiting the proliferation of potential pathogen in sows. Anti-bacteria, anti-oxidative and anti-inflammatory properties ascribed to REE are likely to contribute to performance enhancing effects of sows. We speculate that help to positively modulate the microbiota establishment in their offspring, thus promote hormone secretion and improve health condition of suckling piglets. Whereas, the exact mechanisms underlying these beneficial effects of REE on pigs just as how the REE included in diets of the sows influence piglets' biological responses remain to be verified and further research is required. Even so, the results obtained from our study may be supported by a recent review, reporting the lactocrine hypothesis for maternal programming of reproductive development in pigs. The hypothesis proposed a lactocrine mechanism [1] --- In mammals, maternal effects on development extend into neonatal life. Such effects can be adaptive, conferring benefits to offspring. Therefore, the nutritional basis is presumably that supplementation of the sow diet with REE during late gestation and lactation could increase trophic signals from mother to offspring in colostrum, thereby contributing, at least partially, to a greater litter growth performance during lactation. Ultimately, we have strengthened the discussion and conclusion section on the physiological basis of REE supplementation in sow diet in the revised manuscript (page 14).
Reference:
[1] F. F. Bartol, A. A. Wiley, A. F. George, D. J. Miller and C. A. Bagnell, PHYSIOLOGY AND ENDOCRINOLOGY SYMPOSIUM: Postnatal reproductive development and the lactocrine hypothesis. J Anim Sci, 2017, 95, 2200-2210.
Point 5: Please provide the full composition of REE source. With just La and Ce you are showing no more than 10% of the composition, therefore, can we attributed the results exclusively to this elements?
Response 5: This REE product is a complex chelated with lanthanum, cerium and other inactive carrier materials (diatomite). Diatomite, the low-cost biomaterial with large surface area and biocompatibility, is a good inorganic carrier. It is usually used as drug delivery, so it doesn’t exert any biological function in the body [1]. Thus, we can attribute the results to these rare earth elements. In addition, the characteristic of REE is that low content can play a role, which is consistent with other literature [2,3]. So although the concentration is only 10%, it’s enough to work.
References:
[1] Janićijević J, Krajišnik D, Calija B, et al. Inorganically modified diatomite as a potential prolonged-release drug carrier. Materials Science & Engineering C Materials for Biological Applications, 2014, 42:412-420.
[2] Zhou Q L, Xie J, Ge X P, et al. Growth performance and immune responses of gibel carp, Carassius auratus gibelio, fed with graded level of rare earth-chitosan chelate. Aquaculture International, 2016, 24(2):453-463.
[3] Squadrone S, Stella C, Brizio P, et al. A Baseline Study of the Occurrence of Rare Earth Elements in Animal Feed. Water, Air, & Soil Pollution, 2018, 229(6):190-.
Point 6: How to calculate the coefficient for within-litter birth weight variation (CV)?
Response 6: The coefficient of variation (CV) is the ratio of standard deviation of body weight to average body weight of newborn piglets at birth. Thus, the CV can be used to represent the variation and uniformity of body weight in each litter. The definition has been supplemented to the materials and methods in the revised manuscript and marked as red (page 4).
Point 7: Regarding the “plasma and feces collection”, what was the rationale for choosing just 8 samples/treatments?
Response 7: We performed power calculations using SAS for the experimental results. Our data were considered to reach 80% power at a significant level equal to 0.05, demonstrated that the current number of biological replicates was trustworthy. In addition, for large animals such as sows, there are many researches choose 6 samples, so n = 8 of our study is also statistically persuasive [1,2].
References:
[1] Wan J, Xu Q, He J. Maternal chitosan oligosaccharide supplementation during late gestation and lactation affects offspring growth, Italian Journal of Animal Science, 2018:1-7.
[2] Cindy L B, Ferret-Bernard Stéphanie, Emmanuelle A, et al. Perinatal short-chain fructooligosaccharides program intestinal microbiota and improve enteroinsular axis function and inflammatory status in high-fat diet-fed adult pigs. The FASEB Journal, 2018:fj.201800108R-.
Point 8: The letter and size of figure legends as well as “conclusion and discussion section” need to be revised.
Response 8: We have seriously modified the full text, especially the letter and size of figure legends according to the comments provided by the reviewers in PDF.
Round 2
Reviewer 3 Report
Authors responses are appreciated, however, the experiment was not designed/conducted properly. Groups were not equally distributed (control group (n=60), REE-L(-) group (n=30) and REE-L(+) group (n=30)) and therefore they are not comparable. I suggest to adapt the paper to the "samples" information which is well distributed among groups.
Author Response
Point 1: Authors responses are appreciated, however, the experiment was not designed/conducted properly. Groups were not equally distributed (control group (n=60), REE-L(-) group (n=30) and REE-L(+) group (n=30)) and therefore they are not comparable. I suggest to adapt the paper to the "samples" information which is well distributed among groups.
Response 1: Thank you very much for your rigorous and valuable comments on our manuscript. According to your kind suggestion, we have re-adjusted the animal and experimental design section (page 3) to adapt the paper to the “samples” information which is equally distributed among groups. It is shown as follows:
One hundred and twenty multiparous sows (Landrace × Yorkshire, 3-5 parturition) were selected. The experiment was carried out in two phases, phase one is from late gestation to delivery (G90-G114), sows were comprehensively and equally distributed into two groups: the control group (grain-based diet, basal diet) (n=60) and the REE-G group (the basal diet supplemented with 200 mg REE mixture/kg, REE diet) (n=60). Phase two is from delivery to weaning (L1-L21), there were three groups in this stage: control (n=30), REE-L- (n=30) and REE-L+ (n=30) group. During lactation, sows from REE-G group were further assigned into REE-L-group (REE diet during late gestation + basal diet during lactation) and REE-L+ group (REE diet throughout the late gestation and the lactation). Thirty samples were randomly selected from the control group of phase one using the random number generator in SPSS (SPSS v. 22.0 for Windows; SPSS Inc., Chicago, IL, USA) to make the samples numerically balanced between control and treatment groups. And the control group were continued to be fed a basal diet during the lactation.
At the same time, we re-calculated and re-analysed the data, and presented it in the latest revision. Now in this case, the information coming from “samples” in different treatments is credible.